# Thermal Analysis of Terfenol-D Rods with Different Structures

**DOI:** 10.3390/mi14010216

**Published:** 2023-01-14

**Authors:** Qiang Liu, Xiping He

**Affiliations:** School of Physics and Information Technology, Shaanxi Normal University, Xi’an 710119, China

**Keywords:** Terfenol-D rod, thermal analysis, finite element calculation

## Abstract

To reduce the heating of the Terfenol-D rod and evaluate its working efficiency, six kinds of Terfenol-D rods were designed, and the temperature field of the rods was simulated and calculated using the finite element method to obtain the temperature distribution. The results showed that the untreated rod had the highest temperature; the temperature was higher in the middle and lower at both ends; higher on the outer diameter surface; and lower on the inside. When compared to the untreated rod, the temperatures of sliced rods and slit rods decreased, and the temperature of sliced rods was lower than that of slit rods; the temperature of slit rods was higher in the middle and lower at both ends; the temperature distribution of sliced rods was more uniform relatively; the slice treatment rod had the lowest temperature and the best heat suppression effect. Three structural rods were chosen and manufactured from a total of six that were tested. It shows that the temperature of all rods was higher in the middle and lower at both ends after 30 min of operation. The actual temperature of untreated rod was 34 °C, the actual temperature of radially slit rod was 32 °C, and the actual temperature of sliced rod at both ends was 28 °C. The tested temperature distributions of three rods agreed with the calculated ones.

## 1. Introduction

Ultrasonic technology is widely applied in industry, agriculture, environmental protection, and medicine. As a device to realize electroacoustic energy conversion, the transducer determines the performance of an ultrasonic vibration system. The energy-conversion material of the transducer is mainly divided into piezoelectric ceramic materials and magnetostrictive materials. Compared with piezoelectric ceramic materials, giant magnetostrictive materials (GMM) have a large magnetostrictive coefficient, a high energy density, a fast response speed, and a strong load force. GMM is an excellent smart material for developing high-power and large-amplitude ultrasonic machining systems. It is also widely used in transducers, actuators, and vibration energy collection [1,2,3].

The giant magnetostrictive transducer generates a large amount of heat under the action of a high-frequency alternating magnetic field. The temperature rise will introduce the thermal deformation error and affect the magnetostrictive characteristics of Terfenol-D to make the output unstable; in addition, the temperature rise will cause performance degradation [4]. Zheng [5] studied the output characteristics of the giant magnetostrictive transducer at different temperatures; the temperature rise led to a reduction in the output displacement and the output acoustic sound source level of the transducer. To reduce the ineffective energy loss while cooling the giant magnetostrictive transducer, Liu et al. [6] machined grooves in the axial direction of the magnetic conducting cylinder of the transducer to weaken the temperature rise to some extent. Xie et al. [7] studied the loss and temperature rise characteristics of the giant magnetostrictive actuator, simulated and calculated the temperature distribution of the actuator, and analyzed the factors affecting the working temperature of the Terfenol-D rod. Zhu et al. [8] developed a giant magnetostrictive actuator with a thermal displacement suppression system, which consists of the temperature control module and a thermal displacement compensation module. After the heat transfer mathematical model of the actuator and the heat-induced displacement model of the Terfenol-D rod were built, the heat-induced displacement results were consistent with the results of the heat transfer mathematical model. Zhang et al. [9] established the transient equivalent thermal network model of the giant magnetostrictive transducer and applied the model to estimate the temperature distribution of the transducer. The accuracy of the model was verified by finite element simulations and experiments. Zhou et al. [10] proposed a magnetostrictive jet dispenser with a water cooling system. The experiment shows that the water cooling system can effectively improve the thermal characteristics and uniformity of the magnetostrictive jet dispenser for long-term operation. Zhao et al. [11] proposed a compulsory water cooling control strategy for a giant magnetostrictive actuator’s Terfenol-D constant temperature. The temperature field distribution of the actuator under different input currents and cooling water flow rates was analyzed by the finite element method, and the effectiveness of the temperature control strategy was verified by experiments. Li et al. [12] presented a giant magnetostrictive valve using a tubular Terfenol-D rod as a fluid flow channel and studied the temperature characteristics and output displacement of the giant magnetostrictive valve. 

The results show that the giant magnetostrictive valve was faster than that of the conventional valve, had a stable working temperature range, and provided high precision flow control. Ma et al. [13] presented a novel rotatable giant magnetostrictive transducer in which the excitation coil is fixed and the Terfenol-D rod in the rotary part is excited by the magnetic field; this transducer worked well with much improved ultrasonic vibration stability compared with the traditional transducer. Bai et al. [14] proposed a new design for a giant magnetostrictive transducer with low heat loss. The finite element software was used to design and analyze the internal and external yoke and heat transfer structure, and the results show that the transducer has a low temperature rise. The temperature of the giant magnetostrictive transducer was controlled within an appropriate range; the measures taken included increasing the cooling or heat dissipation system so that it could continue to work steadily, but the excess heat generated reduced the effective power of the transducer. When working in a high-frequency magnetic field, the heating of Terfenol-D rod is the main heat source for the temperature rise of giant magnetostrictive transducers. The research on the heating of Terfenol-D rod is of great significance for the design and optimization of giant magnetostrictive transducers.

He [15] proposed a method to calculate the number of radial slits of the Terfenol-D rod, which reduces the eddy current loss and saves on cost. Teng et al. [16] developed a Terfenol-D transducer; digital slots were set on the Terfenol-D rod to reduce the eddy current loss. Huang et al. [17] cut Terfenol-D into different square rings, studied the changes in its magnetic energy losses under different frequencies and magnetic flux density amplitudes, and analyzed the influence of material anisotropy on the losses. Guo et al. [18] established a high-frequency magnetic energy loss model for giant magnetostrictive materials considering the DC bias magnetic field intensity. The experimental results show that proper DC bias can reduce the losses and improve the output performance of the material. This model has high accuracy compared with the measured results. Li et al. [19] simulated and calculated the eddy current distribution of a Terfenol-D rod with sliced and slit processing using finite element software and studied the vibration performance of a giant magnetostrictive transducer with two kinds of rods. The results show that sliced rods can more effectively suppress eddy currents. Terfenol-D rods were sliced or slit and bonded to reduce eddy current loss.

To improve the efficiency of the giant magnetostrictive transducer and actuator, six structures of Terfenol-D rods were designed in Section 2. The temperature field of Terfenol-D rods with several structures was calculated by the finite element method in Section 3. Three structures of the rods were manufactured, and the temperature of the Terfenol-D rods was tested in Section 4. The paper was concluded in Section 5.

## 2. Six Structures of Terfenol-D Rods

Due to the low resistivity of Terfenol-D, the eddy current loss generated by the material when working in a high-frequency magnetic field was very large; the existence of eddy current increased the energy loss and reduced the driving efficiency of the rod. The rod was cut to make the thickness close to or less than the “penetration depth” to work normally. The limit of the working frequency of the rod was expressed as follows [15]:
(1)$$f_{max}=\frac{1}{\pi \delta_{}^{2}\mu_{0}\mu_{r}\sigma }$$
where *δ* is the thickness, *μ*_0_ is the magnetic permeability of the vacuum, *u_r_* is the relative permeability, and *σ* is the conductivity.

The outer diameter of the Terfenol-D rod was *D* = 18 mm, the inner diameter was *d* = 6 mm, and the length was *l* = 21.5 mm. The working frequency of the rod was 20 kHz, and the skin depth of the rod was 1.2 mm, calculated by (1), which was much smaller than the diameter of the rod. Therefore, the slice thickness was set to 1.2 mm, and the minimum number of slits in the slit processing was calculated according to the working frequency of the rod [15]. The rod was then cut and bonded together, and a 0.4 mm thick epoxy resin layer was poured between the slices and in the slits. Figure 1 shows the six kinds of Terfenol-D rods designed in this way.

## 3. Simulation Calculation and Analysis

There are three distinct modes of heat transfer: heat conduction, thermal convection, and thermal radiation. Heat conduction mainly occurs in Terfenol-D rods. There is thermal convection between the coil and the Terfenol-D rod, and the heat can be taken away if the air flows. Thermal radiation transfer occurs between objects with temperature differences. The heat transfer equation can be expressed as
(2)$$\rho C_{p}\frac{\partial T}{\partial t}+\rho C_{p}\mu \cdot \nabla T+\nabla \left(-k\nabla T\right)=Q$$
*ρ* is the density, *C_P_* is the constant pressure heat capacity, *T* is the temperature, and *k* is the thermal conductivity.

The heat generated by the rod vibrator is determined by the hysteresis and eddy current losses of the Terfenol-D rod and the coil resistance loss. The total heat generated in the rod vibrator can be obtained as: hysteresis loss heat *Q_ml_*, eddy current loss heat *Q_rh_*, and coil resistance loss heat *Q_c_*.

There is a phase angle *β* called the loss angle between the magnetic induction intensity *B* and the magnetic field *H* of Terfenol-D. In an alternating magnetic field, the complex permeability of the Terfenol-D rod is obtained as follows:(3)$$\mu =\frac{1}{\mu_{0}}\frac{B}{H}=\frac{B_{m}}{\mu_{0}H_{m}}e^{-j\beta }=\mu_{r}-j\mu_{i}$$
*μ_r_* is the real permeability and represents the storage of magnetic energy; *μ_i_* is the imaginary permeability and represents the loss of magnetic energy.

During the work of the Terfenol-D rod, the excitation coil provides the required driving magnetic field for the rod, and the wire in the coil will generate resistance loss. The length of the coil can be expressed as follows:(4)$$l_{c}=\pi R_{1}N\left(1+\alpha \right)$$
(5)$$\alpha =\frac{R_{2}}{R_{1}}=2.5$$
*N* is the number of turns of the coil, *R*_1_ is the inner diameter of the coil, and *R*_2_ is the outer diameter of the coil.

The resistance can be written as follows:(6)$$R=\rho \frac{l_{c}}{S}$$
*ρ* is the resistivity of the wire, and *S* is the cross-sectional area of the wire.

The resistance-loss heat of the coil is expressed as follows:(7)$$Q_{c}=I^{2}R$$

The models of several Terfenol-D rods were established in SolidWorks and imported into COMSOL Multiphysics 5.4 for simulation calculations. As shown in Figure 2, the working model of the rod vibrator was established, and the air domain was not shown in the figure.

Table 1 shows the material parameters of the rod vibrator. Using an untreated rod as an example, the total number of volume grid elements was 84,428. The size of the coil was outer diameter *D_c_* = 55 mm, inner diameter *d_c_* = 21 mm, and length *l_c_* = 30 mm; the number of turns was 350; the excitation voltage was 45 V; the operating frequency was 20 kHz; and the heat source was the electromagnetic heat of the Terfenol-D rod and coil. In this work, some parameters, including the wire type, cross-sectional wire area, wire length, number of turns, frequency, and voltage, were assumed to be constants. The magnetic insulation boundary condition was set on the surface of the air domain. The thermal analysis was mainly to determine the boundary conditions of the vibrator, which was called the convective heat transfer coefficient. If no heat dissipation device is added to the vibrator, the heat is dissipated only by natural convection. The natural convective heat transfer coefficient was 15 W/(m^2^·°C), and the ambient temperature was set to 19.5 °C. The frequency-transient study was carried out in COMSOL Multiphysics; the working time was 30 min, and the step length was 1 min. The temperatures of rare earth rods with different structures were obtained from the results.

Figure 3 shows the temperature distribution of Terfenol-D rods with different structures when working for 30 min. It can be seen that the temperature of the untreated rod (Figure 3a) was higher in the middle and lower at both ends; the temperature on the outer diameter surface of the rod was higher than the inner one; and the temperature of the slit rods (Figure 3b,c) was higher in the middle and lower at both ends. However, the temperature distribution of sliced rods (Figure 3d–f) was relatively more uniform.

Figure 4 shows the temperature variation curve of several Terfenol-D rods with time. It can be seen that the temperature of the rods increased with time, and the longer the time was, the less the temperature increased; the untreated rod (Figure 4a) had the highest temperature. Compared with the untreated rod (Figure 4a), the temperatures of the sliced rods (Figure 4d–f) and slit rods (Figure 4b,c) were reduced; the temperature of the sliced rods (Figure 4d–f) was lower than that of the slit rods (Figure 4b,c). The result shows that the sliced and slit rods can reduce heat generation, and the sliced rods can suppress the heating more effectively. Compared to the untreated rod (Figure 4a), the thickness of the material between the two slits in the sliced rods (Figure 4d–f) and slit rods (Figure 4b,c) was reduced, while the thickness of the material between the two slits in the sliced rods (Figure 4d–f) was less than that in the slit rods (Figure 4b,c). The thinner the rod after cutting, the less magnetic energy is lost, and the lower the temperature of the rod. Among the several rods, the slice treatment rod (Figure 4e) has no connecting part between slices, the magnetic energy loss of the rod is the minimum, and the slice treatment rod (Figure 4e) has the lowest temperature and the best heat suppression effect.

## 4. Experimental Test

To further verify the correctness of the simulation, the rods were tested experimentally. Considering the difficulty of bonding the rod after cutting and the processing cost, this work chose to manufacture the radially slit rod (b), the sliced rod at both ends (f), and the untreated rod (a) for the experiment. Figure 5 shows the three structures of Terfenol-D rods.

Figure 6 shows the experimental test diagram of Terfenol-D rods. A high-speed bipolar power supply (BP4620, NF, Japan) was used to supply power to the excitation coil; the applied voltage was 45 V, the DC bias voltage was 0.8 V, and the working frequency was 20 kHz. The axial temperature of the outer diameter surface of the rod was tested by the temperature sensor (type YET-620L, brand KAIPUSEN, China). The temperature sensor had eight temperature probes, which were inserted into the outer diameter surface of the rod from the position of the coil axis and arranged sequentially along the radial direction of the rod.

Figure 7 shows the temperature curve of the Terfenol-D rod with time. It can be seen that the temperature of the untreated rod (Figure 7a) was the highest, and the temperature of the sliced rod at both ends (Figure 7c) was less than that of the radially slit rod (Figure 7b). There is a certain error between the simulation calculation value and the experimental test value of the three types of rods. It may be due to some error between the permeability of the rod used in the simulation calculation and the actual permeability. Therefore, it will have an impact on the actual test results.

Figure 8 shows the axial temperature of three Terfenol-D rods after working for 30 min. It can be seen that the temperature of the three rods was higher in the middle and lower at both ends. Table 2 shows the temperatures of Terfenol-D rods after working for 30 min. The actual temperature of untreated rod was 34.1 °C, the actual temperature of radially slit rod was 32.2 °C, and the actual temperature of sliced rod at both ends was 28.1 °C. Compared to the untreated rod, the actual temperature of the radially slit rod was reduced by 5.88%, and the actual temperature of the sliced rod at both ends was reduced by 17.6%. The relative errors of the simulation calculations and the experimental tests for the temperatures of the three structural rods were (a) 0.19, (b) 0.21, and (f) 0.022, and the simulated temperature trend of the rods agreed with the tested one.

## 5. Conclusions

In this work, six kinds of Terfenol-D rods were studied, and the temperature field of the rods was simulated and calculated by finite element software. As shown from the results, the temperature of the untreated rod was higher in the middle and lower at both ends, higher on the outer diameter surface, and lower on the inside; the temperature of the slit rods was higher in the middle and lower at both ends; the temperature distribution of sliced rods was more uniform. The untreated rod had the highest temperature; compared with the untreated rod, the temperatures of sliced rods and slit rods were reduced, and the temperature of sliced rods was lower than that of slit rods; the slice treatment rod had the lowest temperature among the several rods.

Three structures of rods were manufactured, and the temperature of the rods was tested. After working for 30 min, the actual temperature of untreated rod was 34.1 °C, the actual temperature of radially slit rod was 32.2 °C, and the actual temperature of sliced rod at both ends was 28.1 °C. Compared to the untreated rod, the actual temperature of the radially slit rod was reduced by 5.88%, and the actual temperature of the sliced rod at both ends was reduced by 17.6%. The trends of the numerical calculation values of the temperature of several Terfenol-D rods agreed with the experimentally tested values.

The temperature of the three rods was higher in the middle and lower at both ends; the simulation calculation of the temperature distribution of rods with several structures was basically consistent with the experimental measurement. This research provides an important reference and guidance for the optimization and selection of the rod structure for reducing the heat generation of the giant magnetostrictive transducer and actuator. In the next step, we intend to develop a giant magnetostrictive transducer.

## Figures and Tables

**Figure 1 micromachines-14-00216-f001:**
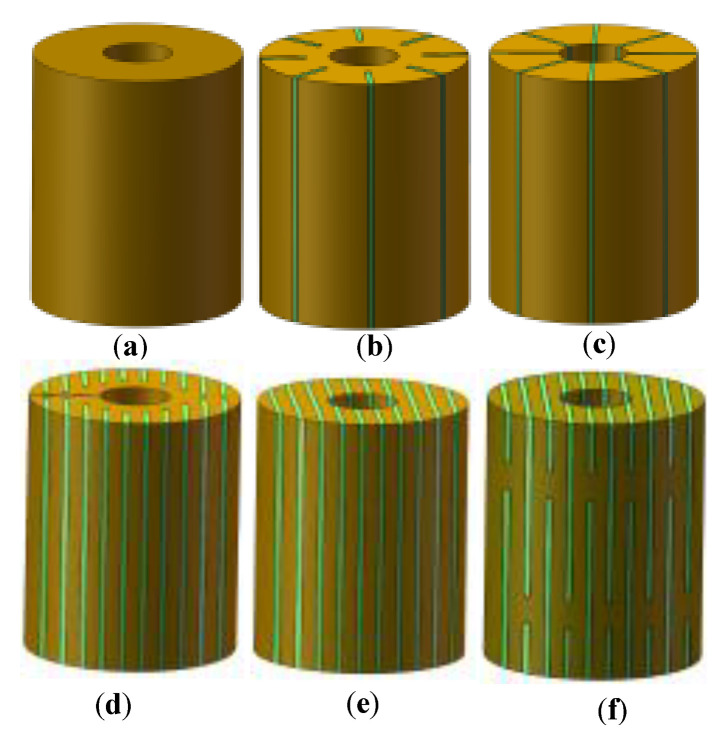
Structural diagram of Terfenol-D rods: (**a**) Untreated; (**b**) Radial slit; (**c**) Radial cut and bonded; (**d**) Sliced and grooved; (**e**) Slice treatment; (**f**) Sliced at both ends.

**Figure 2 micromachines-14-00216-f002:**
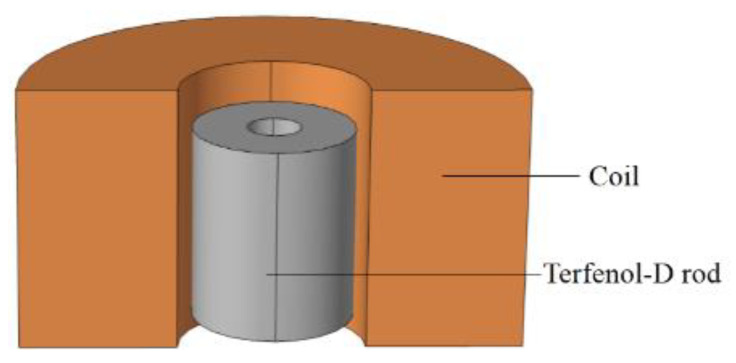
Working model of a rod vibrator.

**Figure 3 micromachines-14-00216-f003:**
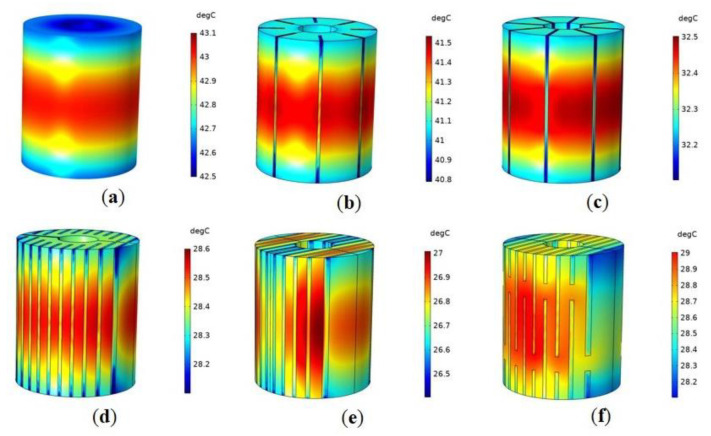
Temperature distribution of Terfenol-D rods with different structures when working for 30 min: (**a**) Untreated; (**b**) Radial slit; (**c**) Radial cut and bonded; (**d**) Sliced and grooved; (**e**) Slice treatment; (**f**) Sliced at both ends.

**Figure 4 micromachines-14-00216-f004:**
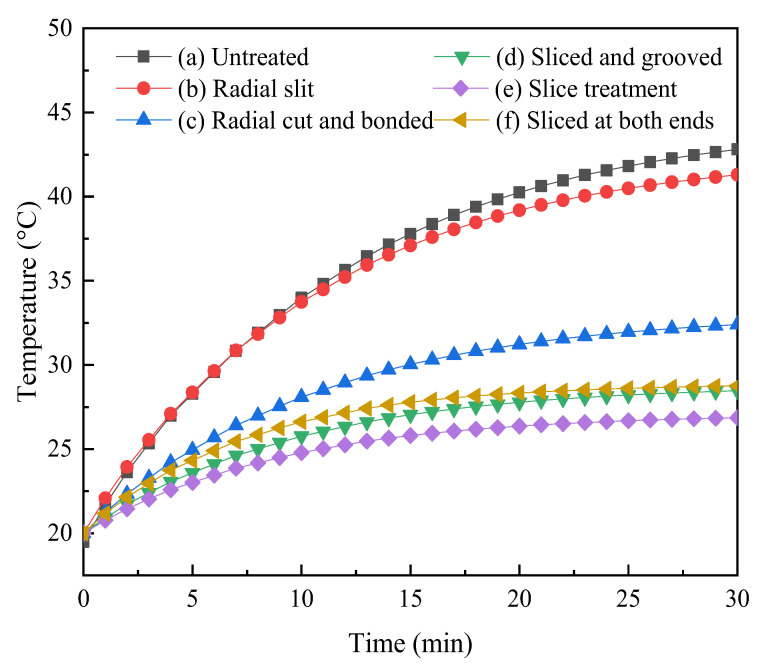
Temperature variation of Terfenol-D rods with time.

**Figure 5 micromachines-14-00216-f005:**
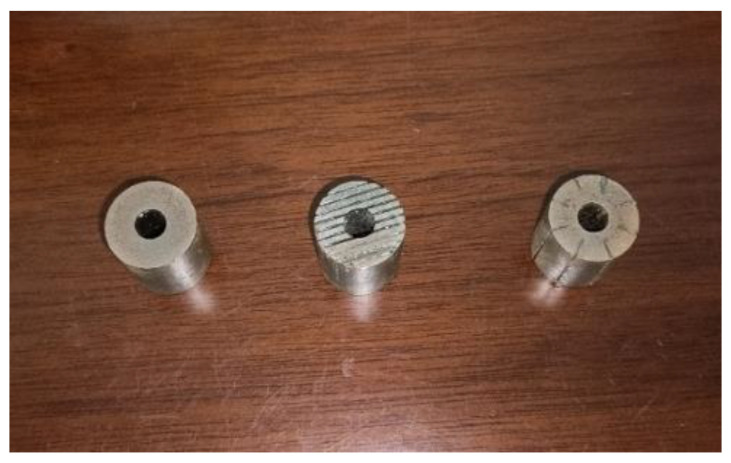
Three structures of Terfenol-D rods for experiment.

**Figure 6 micromachines-14-00216-f006:**
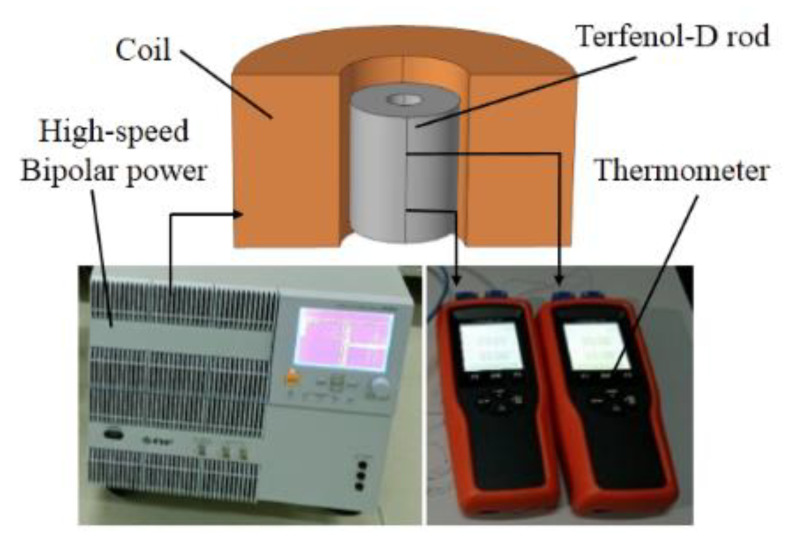
Experimental test diagram of Terfenol-D rods.

**Figure 7 micromachines-14-00216-f007:**
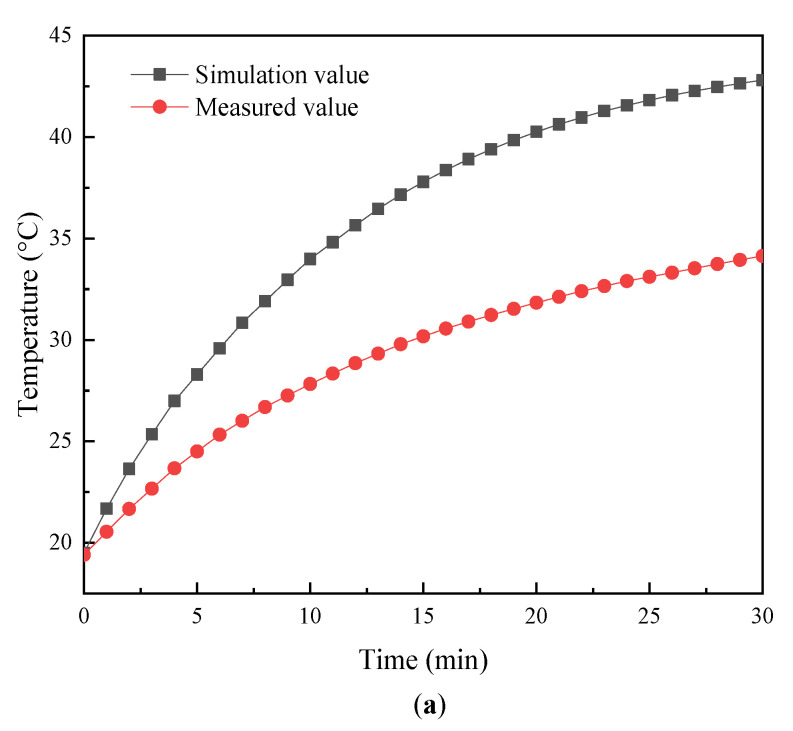
Temperature variation curve of Terfenol-D rods with time: (**a**) untreated; (**b**) radial slit; (**c**) sliced at both ends.

**Figure 8 micromachines-14-00216-f008:**
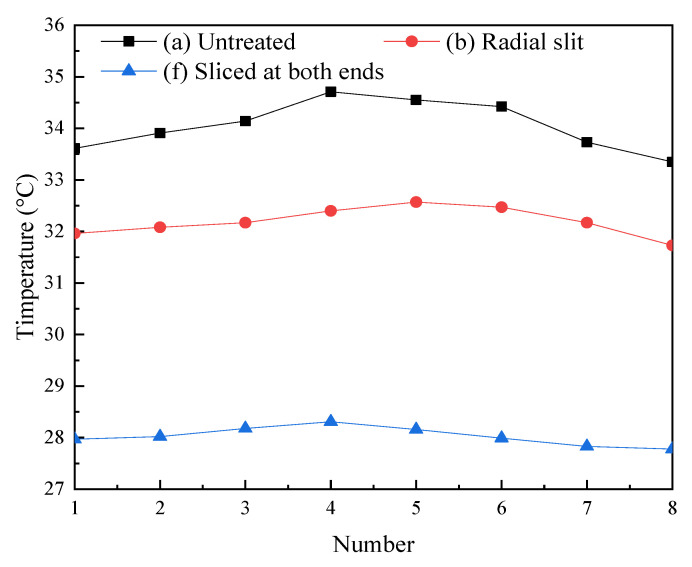
Axial temperature of Terfenol-D rods.

**Table 1 micromachines-14-00216-t001:** Material parameters of rod vibrator.

Name	Relative Permeability	Dielectric Constant	Conductivity (S/m)	Density (kg/m^3^)	Thermal Conductivity (W/m °C)	Specific Heat (J/kg °C)
Terfenol-D	21.9 + 18.2i	1	1.894 × 10^6^	9250	20	200
Coil	1	1	6 × 10^7^	8900	384	394
Epoxy resin	1	3	1	1150	0.2	550

**Table 2 micromachines-14-00216-t002:** Temperatures of Terfenol-D rods after working for 30 min.

Structure of Rods	(a) Untreated	(b) Radial Slit	(c) Radial Cut and Bonded	(d) Sliced and Grooved	(e) Slice Treatment	(f) Sliced at both Ends.
Simulation value/°C	42.8	41.3	32.4	28.5	26.8	28.7
Measured value/°C	34.1	32.2	—	—	—	28.1

## Data Availability

The data that support the findings of this study are available from the corresponding authors upon reasonable request.

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
