# Peer review of "Thermal Analysis of Terfenol-D Rods with Different Structures"

_micromachines, 2023, doi:10.3390/mi14010216_

Round 1
Reviewer 1 Report
Please refer attached review report.

Reviewer 2 Report
Report on the manuscript micromachines-2102719 entitled “Thermal analysis of Terfenol-D rods with different structures”.
The submitted manuscript should be revised. The following points should be addressed
1. Figure 3 should declare the difference between panels, it should have title like the temperature distribution of Terfenol-D rods with different structures when working for 30 minutes.
2. “The result shows that the sliced and slit rods can reduce heat generation, and the sliced rods can suppress the heating more effectively”, the reason should be supported.
3. “there is a certain error between them”, what is the exact value of the error or difference between the measured and simulated data.
Reviewer 3 Report
The manuscript, entitled 'Thermal analysis of Terphenol-D rods with different structures', discusses the temperature distribution for different Terphenol-D rod structures. In the opinion of this reviewer, the manuscript seems interesting and worthy of attention. However, there are some aspects that should be addressed or clarified before further consideration.
1) The state-of-the-art is well addressed, but the novelty of the manuscript and its comparison with the previous ones should be highlighted in a more suitable manner. I kindly request that the authors refer to the novelty of the paper. The improvements achieved with respect to the state of the art have not been clearly pointed out.
2) The organization of the paper should be included at the end of the Introduction section.
3) Please provide more details about the numerical model and computer simulation.
4) Conclusions should be strengthened. The conclusion is to state whether each objective is achieved and what has been achieved. The conclusion part should include any possible impact or benefit of the obtained results.
5) Moreover, the authors should specify which community will benefit from this work and what further research is planned.
6) In the paper there are some minor language errors that should be corrected.
Round 2
Reviewer 1 Report
Authors have included all major comments in revised version. The revised version can be proceed further for acceptance.
Author Response
We have modified the abstract, and the changes are shown in red font

Reviewer 2 Report
The revised version could be accepted
Author Response

(The authors gave the same response as above.)

Reviewer 3 Report
I congratulate the Authors for their research. It is a rich experimental work with several results, written in a direct way.
The comments provided to the first version of the paper have been satisfactorily addressed in the revised version. Thank you.
Author Response

(The authors gave the same response as above.)
